# OpenReview forum: "Reasoner: A Model of Thoughtful Processing Beyond Attention"
_AAAI.org/2025/Workshop/NeurMAD — AAAI 2025 Workshop NeurMAD Submission_

### Official Review · Reviewer_xYBG · 2024-12-19
**reviews for sub9**

**Rating:** 3
**Confidence:** 4

**Review:**

1.it's better to do experiments to give a proof for your methods
2.li's better to make comparison with sota methods.

---

### Official Review · Reviewer_9TEg · 2024-12-20
**Insufficient approach description, missing empirical evidence**

**Rating:** 2
**Confidence:** 4

**Review:**

### Summary
This paper proposes an approach that combines Natural Semantic Metalanguage (NSM) and Bayesian inference to perform structured linguistic reasoning. Input sequences are processed in different steps. First, they are simplified into a predefined set of 65 semantic primes and successively further recombined to represent semantic relationships, as defined in the NSM framework. Then, these basic words and combinations are projected into a vector space, which encodes semantic relations between these vectors. Finally, Bayesian inference is used to generate potential hypotheses about the relationships between concepts and “the nature of the world”. The authors claim that this approach is superior to Transformer-based and neuro-symbolic approaches, showing better contextual understanding, interpretability, and scalability.

### Review
It is impossible to understand from the submitted manuscript how the model really works, as the authors only sketch a vague overview of the different components and do not provide any detail on their practical implementation and integration. No experimental results or empirical evidence are included to support the authors claims. Furthermore, the submission is not anonymous.

---

### Decision · Program_Chairs · 2024-12-30

**Decision:**

Reject

**Comment:**

 We agree with the opinions of  the reviewers.